# Muscle-Specific Effects of Genotype, Animal Age, and Wet Aging Duration on Beef Color, Tenderness, and Sensory Characteristics

**DOI:** 10.3390/ani14243593

**Published:** 2024-12-12

**Authors:** Muhammad Kashif Yar, Muhammad Hayat Jaspal, Sher Ali, Iftikhar Hussain Badar, Muawuz Ijaz, Jibran Hussain

**Affiliations:** 1Department of Meat Science and Technology, University of Veterinary and Animal Sciences, Lahore 54000, Pakistan; sher.ali@uvas.edu.pk (S.A.); iftikhar.hussain@uvas.edu.pk (I.H.B.); 2Department of Animal Sciences, University of Veterinary and Animal Sciences, Jhang Campus, Jhang 35200, Pakistan; muawuz.ijaz@uvas.edu.pk; 3Institute of Food Science and Technology, Sindh Agricultural University Tando Jam, Hyderabad 70050, Pakistan; 4Department of Poultry Production, University of Veterinary and Animal Sciences, Lahore 54000, Pakistan; jibran.hussain@uvas.edu.pk

**Keywords:** aging, bull age, breed, meat quality, muscle-specific

## Abstract

The meat quality, especially color and tenderness, varies significantly among different muscles within a single carcass. Therefore, this study investigated how genotype, animal age, and aging durations affected the beef color, tenderness and sensory characteristics of different muscles. We found that tenderloin and rib-eye muscles improved in color, tenderness, and sensory traits with up to seven days of aging. The sirloin and rump muscles continued to improve for up to 14 days. The tenderloin was the most tender and acceptable, while the sirloin had the best color and oxidative stability. These results highlighted the importance of applying muscle-specific aging strategies to enhance meat quality in both humped and humpless bulls.

## 1. Introduction

Beef color and tenderness are the most dominant quality characteristics affecting the consumers’ acceptance of beef, eating fulfillment, and repeat buying decisions [1]. The bright reddish-purple color indicates the beef’s freshness, and consumers are ready to spend an additional cost for fresh-colored and guaranteed tender beef [2]. However, beef color, tenderness, drip loss, cooking loss, and sensory characteristics are highly variable; muscle-specific characteristics and various muscles from the same carcass exhibit significant differences in meat quality [3]. 

It is reported that the round and chuck muscles are mainly involved in locomotion. Therefore, these muscles are tougher and have less consumer acceptability than rib and loin muscles associated with providing support [3]. Different muscles in a bull carcass show distinct variations in biochemical characteristics such as muscle composition, lipid oxidation [4], and the extent of proteolysis of structural proteins [5]. These variations contribute to intermuscular differences in color, oxidative stability, and beef tenderness [6].

Cattle breeds also affect different beef muscles’ color stability and tenderness [7,8]. Bressan et al. [9] compared the physiochemical characteristics of beef from *Bos taurus* and *Bos indicus* breeds, reporting greater tenderness in *Bos taurus* compared to *Bos indicus* bulls. Miguel et al. [10] reported higher redness in the *Longissimus thoracis* (LT) muscle from Nellore than crossbred (Nellore × Aberdeen Angus) bulls. Animal age at slaughter is another crucial meat quality determinant that affects the color and tenderness of various beef muscles [11]. The color of different beef muscles becomes darker with age, mainly due to increased myoglobin content [12]. Various studies reported that the tenderness of different beef muscles decreases with age, which is largely attributed to an increase in connective tissue along with an increase in their mechanical and thermal stability [11,12,13].

Postmortem wet aging of different meat cuts under the vacuumed packaging condition is a widely adopted practice used to improve the color and tenderness of beef in different regions of the world [3]. The biochemical changes, including the proteolysis of specific structural proteins by endogenous enzymes, are responsible for improving beef quality characteristics during postmortem aging [14]. Traditionally, wet aging is performed on different sub-primal beef cuts; therefore, various muscles in a beef cut undergo a similar aging procedure. However, the beef industry has emphasized marketing single-muscle cuts to provide better beef quality. The muscles differ in physiochemical and quality characteristics; therefore, they might respond differently to postmortem aging.

In South Asia, particularly in Pakistan, where cattle serve as a primary source of red meat, there is no specific beef breed. Instead, beef producers and processors typically classify cattle into humped and humpless categories [15]. These cattle types differ in color, tenderness, oxidative stability, and sensory characteristics [16]. Additionally, early-age slaughtering of calves is a common practice in the region, which adversely affects carcass yield and overall meat quality. Extending the rearing period of these calves by a few months could result in heavier carcasses with better overall meat quality while minimally compromising tenderness. Despite these observations, the impact of genotype, animal age, and aging duration on the meat quality characteristics of *Psoas major* (PM), *Longissimus thoracis* (LT), *Longissimus lumborum* (LL), and *Gluteus Medius* (GM) muscles from humped and humpless cattle remains unexplored. These muscles represent the premium beef cuts, a range of anatomical positions, and functions within a carcass.

This study is innovative as it uniquely evaluated the combined effect of cattle type, animal age, and aging duration on muscle-specific quality characteristics. Furthermore, it explores the potential for these muscles to be marketed individually, which will enhance the quality of these muscles and facilitate the export of these muscles to the higher-end markets of major beef-importing countries. By addressing the research gap in these cattle types and leveraging extended rearing and aging practices, this work would facilitate meeting the growing demand for quality beef. Therefore, the current study aimed to determine the muscle-specific effect of genotype, animal age, and aging duration on the quality characteristics of PM, LT, LL, and GM muscles from humped and humpless cattle bulls.

## 2. Materials and Methods

### 2.1. Animals and Sample Preparation

The meat samples used in the current study were collected from a randomly selected subset of thirty-two animals from our previous research [15]. In summary, a total of *n* = 32 cattle bulls were obtained from a research and development farm, Big Feed Pvt. Ltd., Lahore, Pakistan, comprising 16 humped (*Bos indicus*, hump height ≥ 120 mm) and 16 humpless (*Bos indicus* × *Bos taurus*, hump height ≤ 45 mm) bulls. Each group was further divided by age, with eight bulls from each breed aged 21 ± 2 months and eight bulls from each breed aged 30 ± 3 months. The bulls were housed intensively and raised under similar feeding and management conditions. Ethical approval (Ethic number Dr/451) was obtained from the Institutional Ethical Review Committee, Office of Research Innovation and Commercialization (ORIC), University of Veterinary and Animal Sciences (UVAS), Lahore, Pakistan. The bulls were transported from the farm to a lairage facility at UVAS, Lahore, located 30 km away, with a travel time of approximately 1 h on smooth asphalt roads. Upon arrival, the bulls were rested for 24 h to alleviate transportation stress. During this period, they had unrestricted access to feed and fresh water for the initial 12 h, followed by a 12 h period without feed to minimize cross-contamination from gastrointestinal contents during slaughter. The mean live weight, carcass weight, rib-eye area, and backfat thickness are shown in Appendix A. Following slaughter, the hump height was measured from the carcass by positioning a ruler parallel to the surface of the sawn chine and aligning it perpendicularly to the first thoracic vertebrae. The ruler was moved to the position of the greatest hump width. After 24 h postmortem, four muscles (PM, LT, LL, and GM) were detached from both sides of the beef carcass. Left- and right-sided muscles were divided into one and two sections, alternatively. Each section within each muscle was randomly allocated for either 0, 7, or 14 days of aging at 2 °C. The muscle sections designated for 7 and 14 days of aging were individually placed in nylon–polyethylene bags (90 μm thickness, oxygen transmission rate of 50 cm^3^/m^2^/24 h) and vacuum-sealed using a C100 vacuum packer machine (Multivac^®^ Ltd., Geprüfte Scherhert, AGW, Wolfertschwenden, Germany). Subsequently, the muscle sections underwent wet aging for periods of 7 and 14 days. Following aging, the muscle sections were unwrapped from the vacuum packaging, and each section was divided into four steaks, each 2.50 cm thick: one for color stability; one for thiobarbituric acid reactive substances (TBARS) and drip loss; one for pH, cooking loss, and Warner–Bratzler shear force (WBSF); and one for sensory analysis. Color stability and TBARS were measured for up to seven days of retail display following the completion of the designated aging duration.

### 2.2. pH

The pH of each muscle was assessed using a portable pH meter by directly inserting its probe into the steak. Measurements were taken after 0, 7, and 14 days of aging using a pH meter (WTW, pH 3210-SET 2, Weilheim, Germany). Before use, the pH probe underwent calibration with pH 4.00 and 7.00 buffers at room temperature. Three readings were taken from different locations within each muscle, and results were averaged to perform statistical analysis.

### 2.3. Instrumental Color

After 0, 7, and 14 days of wet aging, the steaks chosen for color measurements were packaged in modified atmosphere trays (HiO_2_ MAP containing 80% O_2_ and 20% CO_2_) using the Multivac^®^ T200 gas packer machine (Multivac^®^ Ltd., Geprufte Scherhert, AGW, Wolfertschwenden, Germany) equipped with a gas mixer (MAP MIX 9001 ME, Dansensor, Ringsted, Denmark). The MAP trays were made from polypropylene, and the MAP film used was PET-PVDC-PE, featuring an oxygen permeability of 5 cm^3^/24 h/m^2^/atm, a carbon dioxide permeability of 20 cm^3^/24 h/m^2^/atm, and a water vapor permeability of 4 g/24 h/m^2^. The steaks were placed under a simulated display for seven days. The color parameters were taken on days 1, 3, 5, and 7 of the simulated retail display using a Minolta chromameter (Konica Minolta^®^ CR-410, Tokyo, Japan), equipped with a 2° standard observer, C illuminant, and 50 mm aperture. The chromameter was calibrated using a standard white tile according to the manufacturer’s instructions. The CIE color parameters, including lightness (L*), redness (a*), yellowness (b*), and chroma (C*, saturation index), were measured by positioning the Minolta head over the MAP trays. Three separate readings were recorded for each steak, and the average of these readings was used for statistical analysis.

### 2.4. Drip Loss (%)

The standard bag method was used to determine the drip loss of the steaks from PM, LT, LL, and GM muscles [17]. The weight of each steak was measured individually with a digital compact weighing balance (SF-400, 7000 g × 1 g, Yongkang, China) and then suspended in a sealed container over 48 h at 4 °C. The drip loss was expressed as a percentage relative to the initial weight of the steak.

### 2.5. Cooking Loss (%) and Instrumental Shear Force

The steaks were weighed using a digital scale (SF-400, 7000 g × 1 g, Yongkang, China) after 0, 7, and 14 days of wet aging, placed in polyethylene bags, and cooked in a water bath maintained at 80 °C. The steaks were cooked in the water bath (Memmert WNB45, Schwabach, Germany) until they reached an internal temperature of 72 °C, monitored with a digital thermometer (TP-101, temperature range of −50 °C to 300 °C, Shanghai, China). After cooking, the steaks were allowed to cool to room temperature (20 °C) for approximately 45 min, and any excess moisture was removed using a hand towel. The steaks were then weighed again. The percentage change in steak weight before and after cooking was used to calculate cooking loss.

To measure shear force, strips of meat with dimensions 5 cm × 1 cm × 1 cm were excised parallel to the muscle fibers using a scalpel blade. These strips were then subjected to shear force perpendicular to the muscle fibers using the ‘V-Slot’ blade on a texture analyzer (TA.XT plus^®^ texture analyzer, Stable Micro System, Godalming, UK). The shear force was recorded in Newtons (N), with five individual readings taken from each steak, and the results were averaged to perform statistical analysis.

### 2.6. Thiobarbituric Acid Reactive Substances (TBARS) Analysis

Lipid oxidation in the meat samples after completing 0, 7, and 14 days of aging was evaluated on days 1, 3, 5, and 7 of the simulated retail displays using the thiobarbituric acid reactive substances (TBARS) method. A mixture was prepared by combining 5 g of ground meat from each steak with 20 mL of distilled water and 25 mL of a 20% aqueous solution of trichloroacetic acid (TCA). Homogenization of the mixture was performed using a Polytron homogenizer (PT 10/35, Brinkman Instruments Inc., Riverview, FL, USA) and was subsequently allowed to incubate at 25 °C for 1 h. The mixture was then centrifuged at 2000 rpm for 15 min, after which the filtrate was diluted to 50 mL with distilled water. Subsequently, 5 mL of the diluted filtrate was combined with 5 mL of a 0.02 M aqueous solution of 2-thiobarbituric acid (TBA) in a test tube. The mixture was incubated in a hot water bath for 15 min to develop color and then cooled in ice water for approximately 10 min. The absorbance of the resulting supernatant was measured at 532 nm using a spectrophotometer (Shimadzu UV-1800, Kyoto, Japan) against a distilled water blank. The TBARS value was expressed as mg of malondialdehyde (MDA) per kg of meat (mg MDA/kg), following the method described by Xiong et al. [18], using the following formula:TBARS value (mg MDA/kg of meat) = 7.8 × absorbance at 532 nm(1)

### 2.7. Sensory Analysis

The trained panelists evaluated the tenderness, juiciness, and flavor of the 0, 7, and 14 days aged steaks from all four (PM, LT, LL, and GM) muscles. A total of 384 beef steaks were randomly evaluated (2 bull types × 2 age groups × 4 muscles × 3 aging times × 8 replicates) during 32 sessions (12 samples in a session and two sessions/day). Thirty-two cooking batches and 12 steaks were randomly selected from a replicate (1 replicate = 2 bull types × 2 age groups × 4 muscles × 3 aging durations). The panelists, consisting of assistant professors, lecturers, and postgraduate students from the Department of Meat Science and Technology, underwent a four-month training program. This program included one-hour sessions held twice a week, totaling 32 h of training. Following this training, a triangle test (ISO 4120:2021) [19] was conducted to select nine panelists capable of discerning variations in tenderness, juiciness levels, and off-flavors in cooked beef steaks. The selected nine panelists evaluated all steaks from all treatments, and the same panelists were used for all sessions. The steaks were cooked on a hot plate until they reached an internal temperature of 72 °C. The steaks were sliced into 1 cm^2^ cubes and randomly served to the panelists at 60 ± 1 °C. The panelists evaluated the steaks using the 9-point hedonic scale for juiciness (9 = extremely juicy and 1 = extremely dry), flavor (9 = full beef flavor and 1 = no beef flavor), tenderness (9 = extremely tender and 1 = not at all tender), and overall acceptability (9 = extremely acceptable and 1 = extremely unacceptable).

### 2.8. Statistical Analysis

The data were analyzed through a linear mixed model using Minitab software (Version 17.3.1). The effects of bull type, age, aging duration, and muscle type were taken as fixed effects, with aging duration as repeated measures and animal as random effects to measure the correlation within samples from the same animals. Mean values were considered significantly different at a significance level of *p* ≤ 0.05. The statistical model used for the analysis is as follows:Y_ijkl_ = μ +F_1i_ + F_2j_ +F_3k_ + F_4l_ + animal_ij_ + ε_ijkl_(2)
where Y_ijkl_ is the response variable, μ is the overall population mean, F_1i_ is the fixed effect of bull type, F_2j_ is the fixed effect of bull age, F_3k_ is the fixed effect of aging duration, F_4l_ is the fixed effect of muscle type, animal_ij_ is the random effect of the individual animal nested within bull type and age, and ε_ijkl_ is the residual error term.

No statistically significant interactions were observed between the fixed effects. Consequently, these interactions were excluded from the tables. Sensory evaluation data for juiciness, flavor, tenderness, and overall acceptability were analyzed using a linear mixed model (PROC GIMMIX), with the fixed effects of bull type, animal age, aging duration, and muscle type along with the random effects of panelist sessions, cooking batches, and individual bulls. The normal distribution of variables was confirmed using histograms and QQ plots. Statistical analysis indicated non-significant differences in random effects, leading to their exclusion from the final model. Tukey–Kramer tests were used for the comparison of mean values, with significance established at *p* ≤ 0.05. Moreover, Pearson correlation analysis was conducted to evaluate the relationships among the various analyzed meat quality characteristics.

## 3. Results and Discussion

### 3.1. pH

The pH of PM, LT, LL, and GM muscles was significantly affected by the bull age and aging duration (*p* < 0.05, Figure 1). All muscles showed a lower (*p* < 0.05) pH in the 30 ± 3 months of age bulls than in the 21 ± 2 months of age bulls. The muscle pH serves as an indicator of glycolysis extent and is also correlated with other meat quality attributes [20]. Older animals generally have higher glycogen reserves [21] and more developed muscles with increased proportions of connective tissue. These factors can influence overall muscle composition and biochemical properties, potentially affecting the ultimate pH. That could be the possible reason for a lower pH in the 30 ± 3 months of age bulls than in the 21 ± 2 months of age bulls. Contrarily, ref. [22] reported an increase in pH values with an increase in age, which might be due to a decrease in glycogen reserves with the age of the bull. In the current study, the pH within all muscles remained higher at 14 days, followed by 7 and 0 days of aging. The increased pH during the aging process could be due to gradual alkalinization induced as a consequence of nitrogenous compound formation during proteolysis [6,23]. Sadowska et al. [24] documented a similar rise in beef pH as the aging duration progressed.

### 3.2. Instrumental Color

The L*, a*, b*, and C* of PM, LT, LL, and GM muscles showed significant variations caused by bull type, age, and aging duration (*p* < 0.05, Table 1, Table 2, Table 3 and Table 4). In the current study, all muscles presented higher (*p* < 0.05) L*, a*, b*, and C* intensities and stabilities in humped bulls than humpless bulls. The higher values of the color intensity and stability of different muscles in humped bulls could be due to lower TBARS values, as reported in the present study. Increased lipid oxidation is known to diminish the intensity and stability of L*, a*, and C* values, while leading to a rise in b* values as a result of heme pigment oxidation. [25]. This is further supported by the correlation analysis, which revealed a strong negative correlation between TBARS and L*, a*, and C* values, while indicating a moderate positive correlation between TBARS and the b* value. (Appendix A). Similarly, Miguel et al. [10] also reported higher L* and a* values in *M. longissimus thoracis* from Nellore cattle (humped) than from the crossbred (Nellore × Aberdeen Angus, humpless) cattle.

In the current study, all muscles showed higher (*p* < 0.05) L*, a*, b*, and C* values in the 30 ± 3 months of age bulls than in the 21 ± 2 months of age bulls. Similarly, Pastsart et al. [20] detected an increase in the color L* and a* values with the increase in the age of bulls. As reported in the current study, the increase in L* values in the bulls aged 30 ± 3 months may be due to their lower pH compared to the bulls aged 21 ± 2 months. This finding is supported by the correlation analysis, which showed a strong negative correlation between pH and L* values (Appendix A). The lower pH causes more protein denaturation and higher L* values in beef [26]. The a* values depend on the myoglobin’s concentration, chemical state, and muscle surface structure and composition. An increase in the a* value in the meat of older bulls was found, which is probably due to an increased amount of myoglobin since its content in the meat increases with age [27].

The postmortem aging duration significantly (*p* < 0.05) affected the L*, a*, b*, and C* values of the PM, LT, LL, and GM muscles during 1, 3, 5, and 7 days of simulated retail display. All the muscles presented higher (*p* < 0.05) L*, a*, b*, and C* values at 14 days, followed by 7 and 0 days of aging during 1, 3, 5, and 7 days of simulated retail display. Various studies reported a rise in the CIE L*, a*, b*, and C* values with the aging duration [28,29,30]. The increase in L* values observed throughout the aging process is likely due to protein denaturation caused by endogenous enzymes, which destabilize the protein matrix and result in increased light scattering [6]. The non-significant difference in the L* values of the PM muscle between 7 and 14 days of aging might be due to a decrease in the protein denaturation (also indicated by WBSF values) that resulted in less light scattering. The changes in the a* values could be attributed to the ability of beef muscles to develop bloom during the aging process. Jacob [31] noted that this effect is linked to a reduction in the activity attributed to the action of oxygen-consuming enzymes and enhanced oxygen penetration, resulting in a thicker oxymyoglobin layer as the aging duration increases.

The color intensity and stability could be affected by the muscle types that have varied rates of oxygen consumption and activities of metmyoglobin reduction [23]. In the present study, LL steaks showed higher while GM and PM steaks showed lower (*p* < 0.05) L*, a*, b*, and C* values measured during simulated retail display (*p* < 0.05, Table 1, Table 2, Table 3 and Table 4). All muscles exhibited a decrease in L*, a*, and C* values, along with an increase in b* values throughout the simulated retail display. However, these changes in meat color values remained within the acceptable range. The differences in the L*, a*, b*, and C* values among PM, LT, LL, and GM can be credited to the muscle fiber differences. The LT and LL muscles composed of type IIB fibers demonstrated greater lightness and redness than the PM and GM muscles composed of type I fibers [32]. The fiber type of LT and LL muscles could favor the glycolytic potential of these muscles and cause rapid postmortem pH decline leading to more protein denaturation and higher L* values due to more light reflectance. The chroma (C*) value indicates the color intensity. Similarly to the current study, Rooyen et al. [33] also found a muscle-specific effect on the C* values of different beef muscles. The PM and GM muscles are considered color labile muscles as these have enhanced mitochondria and a higher oxygen consumption rate. Therefore, these muscles showed lower b* and C* values than the LL and LT muscles.

### 3.3. Drip Loss (%)

The bull age and aging duration significantly affected the drip loss of PM, LT, LL, and GM muscles (*p* < 0.05, Figure 2). All muscles showed higher (*p* < 0.05) drip loss in the 30 ± 3 months of age bulls than in the 21 ± 2 months of age bulls. Similarly, Schönfeldt and Strydom [34] reported a higher drip loss in two or more cattle with permanent incisors than in cattle with no permanent incisors. However, Du Plessis and Hoffman [35] observed no differences in the drip loss of 18 and 30 months of age steers. In the current study, bulls aged 30 ± 3 months exhibited lower pH levels, leading to greater protein denaturation and higher drip losses compared to bulls aged 21 ± 2 months. This observation is further supported by the correlation analysis, which revealed a strong negative correlation between pH and drip loss values (Appendix A). Protein denaturation reduces the water-holding capacity of the meat, causing more water to expel and contributing to higher drip losses.

In the present study, the drip loss of all muscles was higher at 14 days, followed by 7 and 0 days of aging. Similarly, Sadowska et al. [24] also reported higher drip losses up to 21 days of aging in different beef muscles. Moreover, among different muscles, PM and LT showed higher drip loss than LL and GM, but only at 14 days of aging. Likewise, Sadowska et al. [24] also reported higher drip losses in different beef muscles after 14 days of aging. Different muscles have different metabolic and proteolytic potentials [36] that result in differences in meat quality characteristics. PM and LT muscles typically have a different fiber type from LL and GM muscles. PM and LT muscles, which have a higher proportion of fast-twitch fibers [37], are more susceptible to proteolytic activity. This increased proteolysis could be the reason for elevated drip loss.

### 3.4. Cooking Loss (%) and Instrumental Shear Force

The cooking loss displays the water loss from the cooked meat that occurs due to the denaturation of proteins during cooking. The cooking loss of PM, LL, LT, and GM was significantly (*p* < 0.05) affected by the bull type, animal age, and aging duration (Table 5). Only the GM muscle from the humpless bulls showed higher cooking loss than the humped bulls. The increased cooking loss of the GM muscle from humpless bulls could be due to increased proteolysis, which is also evident by the decreased WBSF values in humpless bulls in comparison to humped bulls. Furthermore, the GM muscle in humpless bulls may exhibit unique structural characteristics compared to humped bulls, such as variations in muscle fiber type, connective tissue density, and fat content. These differences in muscle composition contributed to the observed variations in cooking loss [38]. The LL, LT, and GM muscles showed higher (*p* < 0.05) cooking loss in the 30 ± 3 months of age bulls than in the 21 ± 2 months of age bulls. Similarly, Schönfeldt and Strydom [34] observed a rise in cooking loss with increased animal age. A rise in cooking loss with the progression of age might be due to the upsurge of collagen cross-linkages with age, resulting in more cooking loss channels [39].

In the present study, the cooked loss of PM, LL, LT, and GM muscles was significantly increased (*p* < 0.05) with the aging duration. All the muscles showed a higher (*p* < 0.05) cooking loss at 14 days of aging. However, the cooking loss of PM and LT muscles was non-significant between 7 and 14 days of aging. The aging induces an increase in protein denaturation that reduces the proteins’ water-retention ability, consequently increasing cooking losses [30]. However, Hughes et al. [40] found non-significant differences in the cooking loss of beef during aging. During aging, meat undergoes protein denaturation, which can affect its water-holding capacity and cooking loss. In our study, the lack of a significant difference in cooking loss between the PM and LT muscles after 7 and 14 days of aging may be attributed to a stabilization in protein denaturation and an increase in pH following the initial aging period. This stabilization and increase in pH could result in consistent cooking loss values over the subsequent aging days [41]. The GM muscle showed the highest (*p* < 0.05) cooking loss among different muscles in comparison to PM, LT, and LL muscles. Similarly, Rhee et al. [42] reported a higher cooking loss in GM than in the LL muscle. The increased drip loss of GM might be due to the difference in the fat contents [38], as higher marbling is associated with decreased cooking losses.

The shear force of the PM, LT, LL, and GM muscles was significantly (*p* < 0.05) affected by bull type, animal age, and aging duration (Table 5). All the muscles exhibited higher (*p* < 0.05) shear force values in humped bulls than in humpless bulls. The humped bulls have an advanced level of calpastatin activity as compared to the humpless bulls [40]. This increase in calpastatin activity in humped bulls decreases protein proteolysis during aging and negatively impacts meat tenderness [43,44].

The bull age has a significant impact on meat tenderness. The PM, LT, LL, and GM muscles from the 30 ± 3 months of age bulls presented higher (*p* < 0.05) WBSF values than the 21 ± 2 months of age group. The rise in the shear force readings in the 30 ± 3 months of age bulls could be attributed to an increase in connective tissue with the advancement of age. Furthermore, meat becomes tougher due to a rise in collagen’s mechanical and thermal stability with age. As the age of the animal increases, it might result in mature cross-linkages that are more stable and resistant to heat, ultimately leading to less solubilization during the cooking and tougher meat [45].

The postmortem aging time (*p* < 0.05) reduced the WBSF values of the PM, LT, LL, and GM muscles from one to fourteen days of aging. All the muscles showed the lowest (*p* < 0.05) WBSF values at 14 days of aging. However, the PM and LT muscles showed a limited response to aging duration compared to LL and GM, with non-significant changes in WBSF values between 7 and 14 days of aging. This indicated the muscle-specific response of aging duration on the tenderness of different beef muscles. Some muscles, such as LL and *Bicep femoris*, benefit from postmortem aging in terms of tenderness due to their higher proteolytic activity compared to muscles such as PM and *Infraspinatus*, which have limited proteolytic activity [46].

Similarly, Nair et al. [3] also reported a limited response of PM muscle compared to LL muscle to the aging process. Among different beef muscles, PM showed the lowest while GM presented the highest (*p* < 0.05) shear force values. Differences among muscles in the structure, composition, sarcomere length, intermuscular fat, proteolysis rate and extent, and solubility of intramuscular connective tissues contribute to distinct variations in meat tenderness. The present study reported that no single biochemical change could describe the variations in the tenderness of all muscles; therefore, further studies are needed to determine the activity of different proteases and proteomes or any other muscle-specific biochemical change to better evaluate the tenderness of all muscles.

### 3.5. Thiobarbituric Acid Reactive Substances (TBARS)

The TBARS values, indicating lipid oxidation of the PM, LT, LL, and GM muscles were significantly affected by bull type, animal age, and aging duration (*p* < 0.05, Table 6). In the present study, all the muscles presented higher (*p* < 0.05) TBARS values in humpless bulls than in humped bulls. The increased TBARS values in humpless bulls might be due to an increase in unsaturated fatty acid levels in humpless bulls compared to humped bulls [15]. Bull age also affected the (*p* < 0.05) TBARS values. In the current study, all the muscles showed higher (*p* < 0.05) TBARS values in the 30 ± 3 months of age bulls than in the 21 ± 2 months of age bulls. The higher lipid oxidation in elder bulls might be due to increased myoglobin and unsaturated fatty acids compared to younger bulls, as age changes the unsaturation levels in beef lipids [47]. Moreover, lipid oxidation in older bulls is also associated with mitochondrial dysfunction and is due to an increase in the levels of non-heme iron with the age of the bull [48].

The postmortem aging duration significantly (*p* < 0.05) affected the lipid oxidation in PM, LT, LL, and GM muscles. All the muscles showed higher (*p* < 0.05) lipid oxidation at 14 days of aging. However, all the muscles maintained the TBARS values within the maximum acceptable limits of 2.0 mg MDA/kg throughout the aging duration. The increase in lipid oxidation over time could potentially be attributed to the depletion of endogenous antioxidants [49]. Furthermore, iron is released from molecules such as myoglobin and ferritin during the aging process. The amino acids use this iron (produced because of proteolysis during aging) to form the chelates that are the active catalysts of lipid oxidation [50].

Oxidative intensity and stability could be affected by the muscle type and retail display time [51]. The postmortem aging duration significantly (*p* < 0.05) affected the TBARS values of the PM, LT, LL, and GM muscles during 1, 3, 5, and 7 days of simulated display. The LT and LL muscles on day 1 showed the lowest TBARS values while the GM muscle on day 7 showed the highest TBARS values. The difference in the oxidative stability of different muscles could be due to the difference in the muscle fiber type. Muscles such as PM and GM are composed of type I fiber and showed higher mitochondria and oxidative metabolism levels than muscles composed of type IIB such as LT and LL [32].

Moreover, the muscles comprising type I muscle fibers contain higher lipid and myoglobin concentrations than the muscles consisting of type IIB fibers [52]. These differences in muscle fiber types could be the reason for the higher lipid oxidation in PM and GM muscles than in LT and LL muscles. Furthermore, Joseph et al. [53] reported a lower level of the antioxidant protein in the PM muscle than in the LL muscles, which might be another possible reason for the lower oxidative stability in the PM and GM than in the LT and LL muscles in the current study.

### 3.6. Sensory Analysis

The juiciness, flavor, tenderness, and overall acceptability of PM, LL, LT, and GM were significantly (*p* < 0.05) affected by the bull type, age, and aging duration (Table 7). Breed difference is one of the significant features affecting sensory characteristics [43]. In the current study, all the muscles showed better (*p* < 0.05) sensory characteristics in humpless bulls than in humped bulls. The superior juiciness and flavor of humpless bulls may be attributed to differences in marbling levels compared to humped bulls [54]. Furthermore, a decrease in the sensory tenderness in humped bulls could be due to increased calpastatin activity in comparison to the humpless bulls [55]. Bull age at slaughter also affected (*p* < 0.05) the sensory characteristics of the PM, LL, LT, and GM muscles. Juiciness, flavor, and overall acceptability were increased while the sensory tenderness score was decreased in 30 ± 3 months of age bulls in comparison to the 21 ± 2 months of age bulls. An increase in the juiciness, flavor, and overall acceptability score could be due to the increase in the marbling levels with the age of the bull. Furthermore, the quantity and strength of the connective tissue are increased with the advancement of the age of the bull [56], which might result in a decrease in sensory tenderness in the present study. Similarly, Du Plessis et al. [35] observed increased juiciness and a decreased sensory tenderness score in the 30 months of age cattle in comparison to the 18 months of age cattle. Additionally, sensory tenderness also showed a strong negative correlation with WBSF values (Appendix A).

Postmortem aging is commonly used to improve sensory quality characteristics, especially tenderness [57,58]. The 14 days of postmortem aging (*p* < 0.05) improved the juiciness, flavor, tenderness, and overall acceptability of the PM, LL, LT, and GM muscles. The structural and biochemical changes during the aging process improve the juiciness, flavor, tenderness, and overall acceptability [59]. In the present study, the non-significant differences in sensory characteristics between 7 and 14 days of aging duration could be due to a decrease in the structuring and biochemical changes during the mentioned period. Different muscles from the same carcass show considerable variation in sensory characteristics. In the current study, the LT and LL muscles exhibited the highest (*p* < 0.05) juiciness score, while the PM showed the highest flavor, tenderness, and overall acceptability score. The highest juiciness score of the LT and LL muscles might be due to the higher marbling levels in these muscles [38]. Like instrumental tenderness, the PM muscle also showed the highest sensory tenderness score in comparison to the remaining muscles. The highest tenderness of the PM muscle could be the primary reason for the highest overall acceptability score in comparison to the remaining muscles.

## 4. Conclusions

The current study’s findings indicated the muscle-specific effects of animal age and aging duration on the meat quality characteristics of humped and humpless bulls. Humped bulls showed better color and oxidative stability, whereas humpless bulls showed better instrumental tenderness and sensory characteristics. The 30 ± 3 months of age bulls showed better pH color, sensory juiciness, flavor, and an overall acceptability score than the 21 ± 2 months of age bulls. Interestingly, the PM and LT muscles respond differently to the postmortem aging duration compared to the LL and GM muscles. The LL and GM muscles showed improvement in meat tenderness at 14 days of aging, while 7 days of aging improved the tenderness of the PM and LT muscles. The LT and LL muscles showed the highest color and oxidative stability among muscles in comparison to PM and GM. Overall, muscle-specific aging strategies should be used to improve the color, oxidative stability, instrumental tenderness, and sensory characteristics of humped and humpless bulls. Further research could explore additional aging durations and other muscle types to better understand their impact on meat quality characteristics.

## Figures and Tables

**Figure 1 animals-14-03593-f001:**
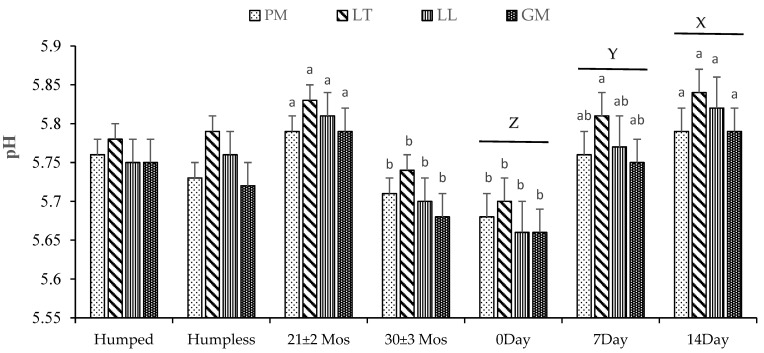
Effects of bull type, age, and wet aging on pH values of *Psoas major* (PM), *Longissimus thoracis* (LT), *Longissimus lumborum* (LL), and *Gluteus Medius* (GM) steaks. a, b: Different alphabets among the same muscle indicated a significant difference (*p* < 0.05) within a treatment. X–Z: Different alphabets among different aging times showed a significant difference (*p* < 0.05). The data were expressed as means ± standard errors.

**Figure 2 animals-14-03593-f002:**
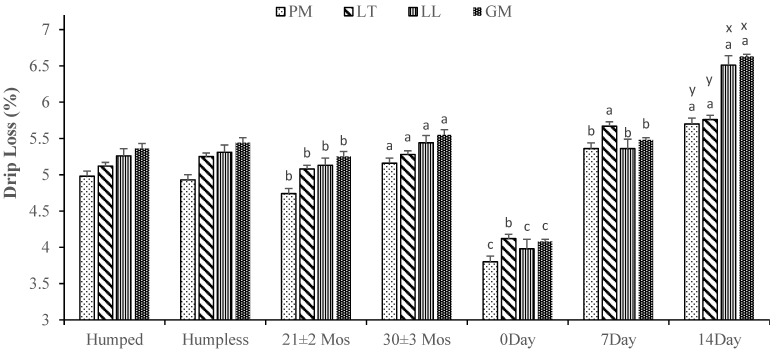
Effects of bull type, age, and wet aging on drip loss (%) of *Psoas major* (PM), *Longissimus thoracis* (LT), *Longissimus lumborum* (LL), and *Gluteus Medius* (GM) steaks. a, b: Different alphabets among the same muscle indicate a significant difference (*p* < 0.05) within a group. x, y: Different alphabets among different muscles indicate a significant difference (*p* < 0.05) within a treatment. The data were expressed as means ± standard errors.

**Table 1 animals-14-03593-t001:** Effects of bull type, age, and wet aging on lightness (L*) values of *Psoas major* (PM), *Longissimus thoracis* (LT), *Longissimus lumborum* (LL), and *Gluteus medius* (GM) steaks during 1, 3, 5, and 7 days of retail display.

Lightness (L*)	Days	Bull Type		Bull Age (Months)			Aging Duration				*p*-Value		
		Humped	Humpless	21 ± 2	30 ± 3	SE	0 Days	7 Days	14 Days	SE	Bull Type	Bull Age	Aging Duration
PM	1	49.21 ^bcde^	48.05 ^bcde^	48.02 ^bcde^	49.24 ^bcde^	0.32	47.71 ^B,bcdef^	48.98 ^AB,abcd^	49.20 ^A,bcdef^	0.39	0.014	0.010	0.022
	3	48.61 ^cdef^	47.25 ^cdef^	47.48 ^cde^	48.41 ^cdef^	0.33	46.87 ^B,cdefg^	48.42 ^A,abcde^	48.54 ^A,cdef^	0.40	0.005	0.047	0.007
	5	47.87 ^def^	46.42 ^efg^	46.68 ^def^	47.62 ^efg^	0.32	46.10 ^B,efgh^	47.69 ^A,bcdef^	47.64 ^A,efgh^	0.39	0.002	0.042	0.008
	7	47.44 ^efg^	46.00 ^fgh^	46.52 ^efg^	46.91 ^fgh^	0.29	45.66 ^B,fgh^	47.31 ^A,cdef^	47.19 ^A,fgh^	0.36	0.001	0.355	0.003
LT	1	51.00 ^ab^	49.71 ^ab^	49.53 ^ab^	51.18 ^ab^	0.39	49.84 ^B,ab^	49.84 ^B,ab^	51.39 ^A,ab^	0.48	0.024	0.005	0.038
	3	50.28 ^abc^	49.01 ^abc^	48.93 ^abc^	50.36 ^abc^	0.38	48.91 ^B,abc^	49.20 ^B,abcd^	50.83 ^A,ab^	0.46	0.022	0.011	0.011
	5	49.62 ^abcd^	48.32 ^abcd^	48.26 ^bcd^	49.67 ^abcde^	0.38	48.37 ^B,abcd^	48.56 ^AB,abcd^	49.97 ^A,abcd^	0.54	0.020	0.011	0.036
	7	49.41 ^bcd^	47.89 ^bcde^	48.05 ^bcde^	49.26 ^bcde^	0.39	48.00 ^B,bcde^	48.29 ^AB,abcde^	49.66 ^A,abcde^	0.48	0.009	0.034	0.042
LL	1	51.40 ^a^	50.18 ^a^	49.95 ^a^	51.64 ^a^	0.39	50.22 ^B,a^	50.30 ^AB,a^	51.85 ^A,a^	0.47	0.033	0.004	0.032
	3	50.68 ^ab^	49.48 ^ab^	49.35 ^ab^	50.82 ^ab^	0.37	49.29 ^B,ab^	49.66 ^B,abc^	51.30 ^A,ab^	0.46	0.030	0.008	0.008
	5	50.01 ^abc^	48.79 ^abc^	48.68 ^abc^	50.13 ^abc^	0.38	48.76 ^B,abc^	49.02 ^AB,abcd^	50.44 ^A,abc^	0.46	0.027	0.009	0.029
	7	49.81 ^abc^	48.37 ^abcd^	48.46 ^abc^	49.72 ^abcd^	0.39	48.38 ^B,abcd^	48.75 ^AB,abcd^	50.13 ^A,abcd^	0.49	0.013	0.028	0.034
GM	1	47.76 ^defg^	46.73 ^defg^	46.71 ^def^	47.79 ^defg^	0.25	46.38 ^B,defgh^	47.22 ^AB,def^	48.14 ^A,defg^	0.31	0.006	0.004	0.001
	3	46.84 ^fgh^	45.67 ^fgh^	45.64 ^fgh^	46.87 ^fgh^	0.29	45.61 ^B,fgh^	46.15 ^AB,efg^	47.00 ^A,fgh^	0.36	0.007	0.005	0.030
	5	45.96 ^gh^	44.98 ^gh^	44.95 ^gh^	45.99 ^gh^	0.30	44.73 ^B,gh^	45.66 ^AB,fg^	46.02 ^A,gh^	0.36	0.023	0.016	0.043
	7	45.43 ^h^	44.18 ^h^	44.36 ^h^	45.25 ^h^	0.28	44.26 ^B,h^	44.47 ^B,g^	45.69 ^A,h^	0.34	0.002	0.026	0.009

PM; *Psoas major*, LT; *Longissimus thoracis*, LL; *Longissimus lumborum*, GM; *Gluteus Medius.* ^A,B^ Means with different superscripts in a row are statistically different (*p* < 0.05). ^a–h^ Means with different superscripts in a column are statistically different (*p* < 0.05).

**Table 2 animals-14-03593-t002:** Effects of bull type, age, and wet aging on redness (a*) values of *Psoas major* (PM), *Longissimus thoracis* (LT), *Longissimus lumborum* (LL), and *Gluteus medius* (GM) steaks during 1, 3, 5, and 7 days of retail display.

Redness (a*)	Days	Bull Type		Bull Age (Months)			Aging Duration				*p*-Value		
		Humped	Humpless	21 ± 2	30 ± 3	SE	0 Days	7 Days	14 Days	SE	Bull Type	Bull Age	Aging Duration
PM	1	21.03 ^bcd^	19.83 ^cd^	19.86 ^cd^	21.00 ^bcde^	0.30	19.18 ^B,cd^	20.96 ^A,cde^	21.16 ^A,cde^	0.37	0.007	0.011	0.001
	3	19.54 ^def^	18.09 ^ef^	18.26 ^ef^	19.36 ^efg^	0.34	17.51 ^B,defg^	19.25 ^A,efg^	19.68 ^A,efg^	0.41	0.004	0.025	0.001
	5	18.55 ^efg^	17.15 f^g^	17.35 ^fg^	18.35 ^gh^	0.35	16.63 ^B,fgh^	18.11 ^A,gh^	18.18 ^A,fg^	0.43	0.007	0.049	0.003
	7	17.81 ^fg^	16.22 ^g^	16.40 ^g^	17.63 ^gh^	0.35	15.81 ^B,h^	17.28 ^AB,h^	17.95 ^A,g^	0.42	0.002	0.015	0.003
LT	1	23.50 ^a^	22.69 ^a^	22.21 ^a^	23.98 ^a^	0.21	22.48 ^B,a^	23.33 ^AB,a^	23.47 ^A,ab^	0.25	0.008	0.000	0.017
	3	22.40 ^ab^	21.49 ^ab^	21.17 ^abc^	22.71 ^ab^	0.25	21.00 ^B,ab^	22.04 ^AB,abc^	22.78 ^A,abc^	0.31	0.013	0.000	0.001
	5	21.50 ^bc^	20.68 ^bc^	20.41 ^bcd^	21.76 ^bcd^	0.27	19.93 ^B,bc^	21.39 ^A,abcd^	21.93 ^A,abcd^	0.33	0.038	0.001	<0.001
	7	20.49 ^cd^	19.65 ^cd^	19.37 ^de^	20.78 ^cdef^	0.28	18.68 ^B,cde^	20.55 ^A,cdef^	20.99 ^A,cde^	0.35	0.040	0.001	<0.001
LL	1	23.60 ^a^	22.44 ^a^	22.20 ^a^	23.85 ^a^	0.23	22.38 ^B,a^	22.95 ^AB,ab^	23.74 ^A,a^	0.28	0.001	<0.001	0.006
	3	22.46 ^ab^	21.35 ^ab^	21.38 ^ab^	22.43 ^abc^	0.25	21.22 ^B,ab^	22.04 ^AB,abc^	22.46 ^A,abcd^	0.30	0.003	0.004	0.018
	5	21.54 ^bc^	20.37 ^bcd^	20.47 ^bcd^	21.43 ^bcd^	0.26	20.09 ^B,bc^	21.21 ^A,bcd^	21.55 ^A,bcde^	0.32	0.003	0.015	0.008
	7	20.46 ^cd^	19.61 ^cd^	19.59 ^de^	20.49 ^def^	0.25	19.10 ^B,cd^	20.37 ^A,cdef^	20.64 ^A,def^	0.31	0.021	0.014	0.002
GM	1	20.27 ^cde^	18.97 ^de^	19.09 ^de^	20.14 ^def^	0.23	18.82 ^B,cd^	19.46 ^B,defg^	20.56 ^A,def^	0.27	<0.001	0.002	0.000
	3	19.46 ^def^	17.98 ^ef^	18.25 ^ef^	19.19 ^fgh^	0.27	17.90 ^B,def^	18.69 ^AB,fgh^	19.57 ^A,efg^	0.33	<0.001	0.016	0.003
	5	18.41 ^fg^	17.30 ^fg^	17.34 f ^g^	18.37 ^gh^	0.28	17.01 ^B,efgh^	17.78 ^AB,gh^	18.76 ^A,fg^	0.34	0.007	0.012	0.003
	7	17.50 ^g^	16.40 ^g^	16.39 ^g^	17.53 ^h^	0.27	15.95 ^B,gh^	17.18 ^A,h^	17.73 ^A,g^	0.33	0.005	0.004	0.001

PM; *Psoas major*, LT; *Longissimus thoracis*, LL; *Longissimus lumborum*, GM; *Gluteus Medius.* ^A,B^ Means with different superscripts in a row are statistically different (*p* < 0.05). ^a–h^ Means with different superscripts in a column are statistically different (*p* < 0.05).

**Table 3 animals-14-03593-t003:** Effects of bull type, age, and wet aging on yellowness (b*) values of *Psoas major* (PM), *Longissimus thoracis* (LT), *Longissimus lumborum* (LL), and *Gluteus medius* (GM) steaks during 1, 3, 5, and 7 days of retail display.

Yellowness (b*)	Days	Bull Type		Bull Age (Months)			Aging Duration				*p*-Value		
		Humped	Humpless	21 ± 2	30 ± 3	SE	0 Days	7 Days	14 Days	SE	Bull Type	Bull Age	Aging Duration
PM	1	10.08 ^h^	8.85 ^i^	8.64 ^g^	10.29 ^g^	0.36	8.49 ^B,h^	9.53 ^AB,h^	10.37 ^A,f^	0.44	0.020	0.002	0.016
	3	10.47 ^gh^	9.37 ^hi^	9.08 ^fg^	10.76 ^fg^	0.35	8.99 ^B,gh^	9.98 ^AB,gh^	10.78 ^A,ef^	0.43	0.032	0.002	0.019
	5	10.82 ^fgh^	9.80 ^ghi^	9.46 ^fg^	11.15 ^efg^	0.36	9.38 ^B,fgh^	10.41 ^AB,fgh^	11.13 ^A,def^	0.43	0.046	0.001	0.021
	7	11.39 ^efgh^	10.37 ^efghi^	9.98 ^efg^	11.78 ^defg^	0.35	9.79 ^B,efgh^	10.99 ^AB,efgh^	11.86 ^A,bcdef^	0.42	0.044	0.001	0.005
LT	1	12.45 ^bcde^	11.70 ^bcdef^	11.46 ^bcd^	12.69 ^bcd^	0.17	11.78 ^B,abcd^	11.87 ^B,bcdef^	12.58 ^A,abcde^	0.21	0.002	0.000	0.016
	3	12.83 ^abcd^	12.13 ^abcd^	11.89 ^bc^	13.07 ^abcd^	0.16	12.08 ^B,abcd^	12.32 ^B,abcde^	13.06 ^A,abcd^	0.20	0.003	<0.001	0.003
	5	13.21 ^abc^	12.53 ^abc^	12.35 ^abc^	13.44 ^abc^	0.15	12.58 ^B,abc^	12.73 ^AB,abcd^	13.38 ^A,abc^	0.33	0.002	<0.001	0.011
	7	13.67 ^ab^	12.82 ^ab^	12.75 ^ab^	13.74 ^ab^	0.15	12.93 ^B,ab^	13.17 ^AB,ab^	13.63 ^A,ab^	0.35	<0.001	0.001	0.038
LL	1	11.94 ^cdef^	11.24 ^cdefg^	11.00 ^cde^	12.18 ^cdef^	0.16	11.19 ^B,bcde^	11.43 ^B,cdefg^	12.15 ^A,bcdef^	0.20	0.003	<0.001	0.003
	3	12.57 ^bcde^	11.89 ^bcde^	11.72 ^bcd^	12.74 ^bcd^	0.15	11.76 ^B,abcd^	12.12 ^B,bcde^	12.81 ^A,abcd^	0.18	0.002	0.001	0.001
	5	13.33 ^abc^	12.67 ^abc^	12.57 ^ab^	13.42 ^abc^	0.14	12.44 ^B,abc^	13.05 ^A,abc^	13.49 ^A,ab^	0.17	0.001	<0.001	<0.001
	7	14.17 ^a^	13.60 ^a^	13.50 ^a^	14.27 ^a^	0.14	13.40 ^B,a^	13.94 ^AB,a^	14.29 ^A,a^	0.18	0.007	0.001	0.005
GM	1	10.84 ^fgh^	9.80 ^ghi^	9.48 ^fg^	11.16 ^efg^	0.35	9.38 ^B,fgh^	10.41 ^AB,fgh^	11.17 ^A,def^	0.43	0.042	0.001	0.018
	3	11.26 ^efgh^	10.23 ^fghi^	9.89 ^efg^	11.59 ^defg^	0.34	9.86 ^B,efgh^	10.87 ^AB,efgh^	11.50 ^A,cdef^	0.42	0.038	0.001	0.027
	5	11.91 ^defg^	10.69 ^defgh^	10.47 ^def^	12.13 ^cdef^	0.33	10.46 ^B,defg^	11.30 ^AB,defg^	12.15 ^A,bcdef^	0.34	0.012	0.001	0.018
	7	12.41 ^bcde^	11.23 ^cdefg^	11.07 ^cde^	12.56 ^bcde^	0.33	11.05 ^B,cdef^	11.75 ^AB,bcdef^	12.65 ^A,abcde^	0.33	0.014	0.002	0.024

PM; *Psoas major*, LT; *Longissimus thoracis*, LL; *Longissimus lumborum*, GM; *Gluteus Medius.* ^A,B^ Means with different superscripts in a row are statistically different (*p* < 0.05). ^a–i^ Means with different superscripts in a column are statistically different (*p* < 0.05).

**Table 4 animals-14-03593-t004:** Effects of bull type, age, and wet aging on chroma (C*) values of *Psoas major* (PM), *Longissimus thoracis* (LT), *Longissimus lumborum* (LL), and *Gluteus medius* (GM) steaks during 1, 3, 5, and 7 days of retail display.

Chroma (C*)	Days	Bull Type		Bull Age (Months)			Aging Duration				*p*-Value		
		Humped	Humpless	21 ± 2	30 ± 3	SE	0 Days	7 Days	14 Days	SE	Bull Type	Bull Age	Aging Duration
PM	1	22.37 ^cde^	21.09 ^def^	20.73 ^cde^	22.74 ^bcde^	0.36	20.78 ^B,cd^	21.87 ^AB,cdef^	22.50 ^A,cdef^	0.45	0.017	<0.001	0.029
	3	20.56 ^efg^	19.25 ^gh^	18.99 ^fg^	20.83 ^efgh^	0.39	18.83 ^B,ef^	19.87 ^AB,fghi^	21.02 ^A,efgh^	0.47	0.021	0.002	0.008
	5	19.76 ^fg^	18.19 ^hi^	18.36 ^gh^	19.59 ^ghi^	0.42	17.77 ^B,fg^	18.93 ^AB,hi^	20.22 ^A,fgh^	0.52	0.011	0.044	0.006
	7	18.96 ^g^	17.13 ^i^	17.35 ^h^	18.74 ^i^	0.44	16.84 ^B,g^	18.33 ^AB,i^	18.96 ^A,h^	0.53	0.005	0.028	0.021
LT	1	24.86 ^ab^	23.78 ^ab^	23.13 ^ab^	25.51 ^a^	0.26	23.44 ^B,a^	24.45 ^AB,ab^	25.07 ^A,ab^	0.32	0.006	<0.001	0.003
	3	23.19 ^bcd^	22.16 ^cde^	21.81 ^bcd^	23.54 ^bc^	0.30	21.26 ^B,bc^	23.08 ^A,abcd^	23.68 ^A,abcd^	0.36	0.019	<0.001	<0.001
	5	22.51 ^cde^	21.61 ^def^	21.27 ^cde^	22.84 ^bcd^	0.30	20.63 ^B,cde^	22.45 ^A,bcde^	23.09 ^A,bcde^	0.37	0.038	0.001	<0.001
	7	21.59 ^def^	20.65 ^efg^	20.46 d^ef^	21.79 ^cdef^	0.31	19.68 ^B,cde^	21.41 ^A,defg^	22.29 ^A,cdef^	0.38	0.037	0.004	<0.001
LL	1	25.52 ^a^	24.24 ^a^	23.94 ^a^	25.83 ^a^	0.25	24.30 ^B,a^	24.79 ^AB,a^	25.55 ^A,a^	0.32	0.001	<0.001	0.026
	3	24.31 ^abc^	23.30 ^abc^	23.32 ^ab^	24.29 ^ab^	0.28	23.15 ^B,ab^	23.67 ^AB,abc^	24.60 ^A,abc^	0.34	0.013	0.018	0.014
	5	23.16 ^bcd^	22.28 ^bcd^	22.25 ^bc^	23.19 ^bc^	0.28	21.37 ^C,bc^	22.69 ^B,abcd^	24.09 ^A,abcd^	0.35	0.034	0.024	<0.001
	7	22.23 ^de^	21.42 ^def^	21.38 ^cde^	22.27 ^cde^	0.27	20.28 ^C,cde^	21.92 ^B,cdef^	23.26 ^A,abcde^	0.31	0.029	0.017	<0.001
GM	1	21.56 ^def^	20.95 ^def^	20.70 ^cde^	21.81 ^cdef^	0.20	20.66 ^B,cde^	21.03 ^B,defg^	22.07 ^A,def^	0.25	0.038	<0.001	0.001
	3	21.09 ^ef^	20.26 ^fg^	20.11 ^ef^	21.24 ^defg^	0.24	20.19 ^B,cde^	20.50 ^AB,efgh^	21.33 ^A,efg^	0.30	0.019	0.002	0.027
	5	20.17 ^fg^	19.11 ^gh^	19.03 ^fg^	20.24 ^fghi^	0.27	18.94 ^B,def^	19.71 ^AB,ghi^	20.26 ^A,fgh^	0.33	0.007	0.002	0.022
	7	18.91 ^g^	18.14 ^hi^	17.99 ^gh^	19.06 ^hi^	0.25	17.67 ^B,fg^	18.52 ^AB,hi^	19.38 ^A,gh^	0.31	0.033	0.004	0.001

PM; *Psoas major*, LT; *Longissimus thoracis*, LL; *Longissimus lumborum*, GM; *Gluteus Medius*, ^A–C^ Means with different superscripts in a row are statistically different (*p* < 0.05). ^a–i^ Means with different superscripts in a column are statistically different (*p* < 0.05).

**Table 5 animals-14-03593-t005:** Effects of bull type, age, and wet aging on cooking loss (%) and Warner–Bratzler shear force (WBSF) values of *Psoas major* (PM), *Longissimus thora*cis (LT), *Longissimus lumborum* (LL), and *Gluteus medius* (GM) steaks.

Parameter	Muscle	Bull Type		Bull Age (Months)			Aging Duration				*p*-Value		
		Humped	Humpless	21 ± 2	30 ± 3	SE	0 Days	7 Days	14 Days	SE	Bull Type	Bull Age	Aging Duration
Cooking Loss	PM	31.76 ^b^	32.00 ^b^	31.58 ^b^	32.18 ^b^	0.28	30.18 ^B,b^	32.53 ^A,b^	32.93 ^A,b^	0.34	0.541	0.129	<0.001
(%)	LT	31.63 ^b^	31.87 ^b^	31.22 ^b^	32.28 ^b^	0.30	29.85 ^B,b^	32.10 ^A,b^	33.30 ^A,b^	0.37	0.562	0.017	<0.001
	LL	31.70 ^b^	32.29 ^b^	31.42 ^b^	32.58 ^b^	0.10	29.94 ^C,b^	32.32 ^B,b^	33.73 ^A,ab^	0.27	0.066	0.001	<0.001
	GM	33.22 ^a^	34.01 ^a^	33.19 ^a^	34.05 ^a^	0.24	31.38 ^C,a^	34.23 ^B,a^	35.24 ^A,a^	0.30	0.027	0.015	<0.001
WBSF	PM	33.43 ^b^	31.03 ^b^	30.83 ^c^	33.63 ^b^	0.78	37.13 ^A,c^	30.66 ^B,c^	28.91 ^B,c^	0.96	0.036	0.016	<0.001
(N/cm^2^)	LT	41.78 ^a^	39.00 ^a^	38.22 ^b^	42.55 ^a^	0.53	48.57 ^A,b^	37.12 ^B,b^	35.47 ^B,b^	0.65	0.001	<0.001	<0.001
	LL	44.63 ^a^	42.64 ^a^	42.13 ^ab^	45.14 ^a^	0.48	51.21 ^A,ab^	42.87 ^B,a^	36.82 ^C,a^	0.58	0.005	<0.001	<0.001
	GM	45.12 ^a^	43.05 ^a^	42.74 ^a^	45.42 ^a^	0.24	52.29 ^A,a^	44.09 ^B,a^	35.86 ^C,a^	0.30	0.003	<0.001	<0.001

PM; *Psoas major*, LT; *Longissimus thoracis*, LL; *Longissimus lumborum*, GM; *Gluteus Medius*, WBSF; Warner–Bratzler shear force. ^A–C^ Means with different superscripts in a row are statistically different (*p* < 0.05). ^a–c^ Means with different superscripts in a column are statistically different (*p* < 0.05).

**Table 6 animals-14-03593-t006:** Effects of bull type, age, and wet aging on thiobarbituric acid reactive substances (TBARS) values of *Psoas major* (PM), *Longissimus thoracis* (LT), *Longissimus lumborum* (LL), and *Gluteus medius* (GM) steaks during 1, 3, 5, and 7 days of retail display.

TBARS (mg MDA/kg)	Days	Bull Type		Bull Age (Months)			Aging Duration				*p*-Value		
		Humped	Humpless	21 ± 2	30 ± 3	SE	0 Days	7 Days	14 Days	SE	Bull Type	Bull Age	Aging Duration
PM	1	0.89 ^cde^	1.04 ^bcdef^	0.90 ^cdefg^	1.03 ^bcde^	0.04	0.73 ^C,bcd^	0.95 ^B,cdef^	1.21 ^A,bcdefg^	0.05	0.008	0.011	<0.001
	3	0.94 ^bcde^	1.10 ^abcde^	0.95 ^bcdef^	1.08 ^abcde^	0.04	0.79 ^C,abcd^	0.99 ^B,bcdef^	1.27 ^A,abcdef^	0.04	0.004	0.016	<0.001
	5	0.99 ^abcd^	1.14 ^abcde^	1.00 ^abcde^	1.12 ^abcd^	0.03	0.83 ^C,abc^	1.05 ^B,abcde^	1.31 ^A,abcd^	0.04	0.006	0.020	<0.001
	7	1.04 ^abc^	1.19 ^abc^	1.06 ^abcd^	1.18 ^abc^	0.04	0.88 ^C,abc^	1.11 ^B,abc^	1.36 ^A,abc^	0.04	0.006	0.022	<0.001
LT	1	0.72 ^e^	0.94 ^def^	0.75 ^fg^	0.93 ^de^	0.05	0.71 ^B,bcd^	0.83 ^AB,ef^	0.98 ^A,h^	0.07	0.002	0.023	0.019
	3	0.86 ^cde^	1.08 ^abcdef^	0.84 ^defg^	1.10 ^abcde^	0.05	0.84 ^B,abc^	0.95 ^AB,cdef^	1.11 ^A,efgh^	0.06	0.004	0.001	0.017
	5	0.95 ^abcd^	1.14 ^abcde^	0.91 ^bcdefg^	1.19 ^abcd^	0.05	0.92 ^B,ab^	1.04 ^AB,bcdef^	1.18 ^A,cdefgh^	0.07	0.016	0.000	0.024
	7	1.04 ^abc^	1.22 ^abcd^	1.01 ^abcde^	1.25 ^abc^	0.04	1.02 ^B,a^	1.07 ^B,abcde^	1.30 ^A,abcdef^	0.06	0.010	0.001	0.003
LL	1	0.71 ^e^	0.85 ^f^	0.71 ^g^	0.84 ^e^	0.03	0.57 ^C,d^	0.78 ^B,f^	0.97 ^A,gh^	0.04	0.002	0.003	<0.001
	3	0.79 ^de^	0.91 ^ef^	0.78 ^efg^	0.92 ^de^	0.03	0.67 ^C,cd^	0.85 ^B,def^	1.03 ^A,fgh^	0.03	0.004	0.001	<0.001
	5	0.85 ^cde^	0.97 ^cdef^	0.84 ^defg^	0.98 ^cde^	0.03	0.71 ^C,bcd^	0.92 ^B,cdef^	1.10 ^A,defgh^	0.03	0.003	0.001	<0.001
	7	0.94 ^bcde^	1.04 ^abcdef^	0.92 ^bcdefg^	1.06 ^abcde^	0.02	0.82 ^C,abc^	0.99 ^B,bcdef^	1.16 ^A,bcdefgh^	0.03	0.004	0.000	<0.001
GM	1	1.02 ^abcd^	1.12 ^abcde^	1.00 ^abcd^	1.13 ^abcd^	0.03	0.81 ^C,abc^	1.10 ^B,abcd^	1.30 ^A,abcde^	0.04	0.020	0.010	<0.001
	3	1.07 ^abc^	1.18 ^abcd^	1.08 ^abc^	1.18 ^abc^	0.03	0.87 ^C,abc^	1.15 ^B,abc^	1.36 ^A,abc^	0.04	0.016	0.019	<0.001
	5	1.13 ^ab^	1.22 ^ab^	1.12 ^ab^	1.23 ^ab^	0.03	0.91 ^C,ab^	1.21 ^B,ab^	1.40 ^A,ab^	0.03	0.020	0.013	<0.001
	7	1.17 ^a^	1.28 ^a^	1.18 ^a^	1.29 ^a^	0.03	0.97 ^C,a^	1.27 ^B,a^	1.45 ^A,a^	0.04	0.015	0.015	<0.001

PM; *Psoas major*, LT; *Longissimus thoracis*, LL; *Longissimus lumborum*, GM; *Gluteus Medius*, TBARS; thiobarbituric acid reactive substances. ^A–C^ Means with different superscripts in a row are statistically different (*p* < 0.05). ^a–h^ Means with different superscripts in a column are statistically different (*p* < 0.05).

**Table 7 animals-14-03593-t007:** Effects of bull type, age, and wet aging on sensory evaluations of *Psoas major* (PM), *Longissimus thoracis* (LT), *Longissimus lumborum* (LL), and *Gluteus medius* (GM) steaks.

Sensory Evaluation	Muscle	Bull Type		Bull Age (Months)			Aging Duration				*p*-Value		
		Humped	Humpless	21 ± 2	30 ± 3	SE	0 Days	7 Days	14 Days	SE	Bull Type	Bull Age	Aging Duration
Juiciness	PM	5.90 ^b^	6.67 ^b^	5.98 ^b^	6.60 ^b^	0.13	5.20 ^B,b^	6.65 ^A,b^	7.00 ^A,bc^	0.16	<0.001	0.001	<0.001
	LT	7.00 ^a^	7.83 ^a^	7.13 ^a^	7.70 ^a^	0.14	6.65 ^B,a^	7.65 ^A,a^	7.95 ^A,a^	0.17	<0.001	0.005	<0.001
	LL	6.80 ^a^	7.70 ^a^	6.93 ^a^	7.57 ^a^	0.13	6.60 ^B,a^	7.40 ^A,a^	7.75 ^A,ab^	0.16	<0.001	0.001	<0.001
	GM	5.67 ^b^	6.43 ^b^	5.73 ^b^	6.37 ^b^	0.13	5.10 ^C,b^	6.20 ^B,b^	6.85 ^A,c^	0.15	<0.001	0.001	<0.001
Flavor	PM	7.23 ^a^	7.73 ^a^	7.27 ^a^	7.70 ^a^	0.12	6.65 ^B,a^	7.75 ^A,a^	8.05 ^A,a^	0.15	0.005	0.013	<0.001
	LT	6.33 ^b^	7.03 ^b^	6.50 ^a^	6.87 ^b^	0.12	5.65 ^B,b^	7.05 ^A,b^	7.35 ^A,b^	0.15	<0.001	0.039	<0.001
	LL	6.53 ^ab^	7.27 ^ab^	6.57 ^a^	7.23 ^ab^	0.13	5.95 ^B,ab^	7.25 ^A,ab^	7.50 ^A,ab^	0.16	<0.001	0.001	<0.001
	GM	7.16 ^a^	7.73 ^a^	7.13 ^a^	7.77 ^a^	0.16	6.55 ^B,a^	7.85 ^A,a^	7.95 ^A,ab^	0.20	0.018	0.009	<0.001
Tenderness	PM	7.17 ^a^	7.83 ^a^	7.77 ^a^	7.23 ^a^	0.13	6.65 ^B,a^	7.80 ^A,a^	8.05 ^A,a^	0.16	0.001	0.005	<0.001
	LT	6.80 ^a^	7.63 ^a^	7.53 ^a^	6.90 ^a^	0.13	6.60 ^B,a^	7.40 ^A,a^	7.65 ^A,a^	0.16	<0.001	0.001	<0.001
	LL	7.00 ^a^	7.50 ^a^	7.53 ^a^	6.97 ^a^	0.17	6.40 ^B,a^	7.40 ^A,a^	7.95 ^A,a^	0.21	0.043	0.023	<0.001
	GM	5.53 ^b^	6.23 ^b^	6.17 ^b^	5.60 ^b^	0.14	5.00 ^B,b^	6.20 ^A,b^	6.45 ^A,b^	0.17	0.001	0.004	<0.001
Overall	PM	7.00 ^a^	7.63 ^a^	6.87 ^a^	7.77 ^a^	0.20	6.65 ^B,a^	7.45 ^AB,a^	7.85 ^A,a^	0.28	0.027	0.002	0.003
Acceptability	LT	5.83 ^b^	6.57 ^b^	5.90 ^b^	6.50 ^b^	0.18	5.65 ^B,ab^	6.30 ^AB,b^	6.65 ^A,b^	0.22	0.005	0.020	0.007
	LL	5.87 ^b^	6.80 ^b^	6.03 ^b^	6.63 ^b^	0.17	5.45 ^C,b^	6.35 ^B,b^	7.20 ^A,ab^	0.20	<0.001	0.013	<0.001
	GM	5.67 ^b^	6.37 ^b^	5.77 ^b^	6.27 ^b^	0.17	5.20 ^C,b^	6.05 ^B,b^	6.80 ^A,b^	0.20	0.004	0.037	<0.001

PM; *Psoas major*, LT; *Longissimus thoracis*, LL; *Longissimus lumborum*, GM; *Gluteus Medius.* ^A–C^ Means with different superscripts in a row are statistically different (*p* < 0.05). ^a–c^ Means with different superscripts in a column are statistically different (*p* < 0.05).

## Data Availability

The original contributions presented in this study are included in the article/Appendix A. Further inquiries can be directed to the corresponding authors.

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
