# Peer review of "Muscle-Specific Effects of Genotype, Animal Age, and Wet Aging Duration on Beef Color, Tenderness, and Sensory Characteristics"

_animals, 2024, doi:10.3390/ani14243593_

Round 1
Reviewer 1 Report (Previous Reviewer 2)
Comments and Suggestions for Authors
L22 muscle type or breed type?
1 Introduction. There are already many similar studies have been reported, what are the innovation and highlight of this research? Why these two types of bulls, and two ages were selected in this study? This is not reflected in the background introduction, so suggest highlighting the innovative points of this study.
L204-205 aging duration was repeated here, is ageing duration belonged to fixed effects or random effects? Please clarify this point.
3. Results and Discussion & 4 Conclusions. Is the description of the results and conclusions correct? Please must carefully check all results and make sure the accuracy of the description of results and conclusion. For example:
In conclusion, it’s stated that ‘’humpless bulls showed better instrumental tenderness and sensory characteristics’’, but from Table 5, there is no significant difference between Humped bulls and Humpless bulls; from Table 7, there is also no significant difference between two bull types, so how do you find those result and conclusion?
Similarly, In conclusion, it’s stated that ‘’All meat quality characteristics were improved in 30±3 months than 21±2 months of age bulls’’. Why can you say this? I see from Table 5, this is no significant difference of cooking loss between two ages, and for WBSF, 30±3 months’ age bulls have higher value than those of 21±2 months.
Author Response
Response to Reviewer # 1
Comment: L22 muscle type or breed type?
Reply: Thank you for highlighting important aspects. The sentence has been corrected and mentioned in line 22 of the revised manuscript.
Comment: Introduction. There are already many similar studies have been reported, what are the innovation and highlight of this research? Why these two types of bulls, and two ages were selected in this study? This is not reflected in the background introduction, so suggest highlighting the innovative points of this study.
Reply: Thank you for pointing out the importance of highlighting the novelty and research gap of this study, particularly regarding the selection of cattle types and animal age.
In South Asia, particularly in Pakistan, cattle are a primary source of red meat; however, there are no specific beef breeds in the region. Instead, beef producers and processors categorize cattle into humped and humpless types, which exhibit notable differences in color, tenderness, oxidative stability, and sensory characteristics. Moreover, the practice of early-age slaughtering of calves in this region is widespread, which negatively impacts carcass yield and overall meat quality. Extending the rearing period of calves by a few months has the potential to produce heavier carcasses with improved meat quality while only minimally compromising tenderness.
Despite these observations, to the best of our knowledge, no prior studies have examined the combined effects of genotype, animal age, and aging duration on the meat quality characteristics of the Psoas major (PM), Longissimus thoracis (LT), Longissimus lumborum (LL), and Gluteus medius (GM) muscles in humped and humpless cattle. These muscles represent premium beef cuts and vary in anatomical position and function, making their study particularly significant.
In response to your suggestion, we have thoroughly revised the introduction section to explicitly address the study’s novelty and research gap, with a specific focus on the selection of cattle types and ages. The revised content can be found in lines 73–92 of the manuscript and is highlighted in red for your convenience.
Comment: L204-205 aging duration was repeated here, is ageing duration belonged to fixed effects or random effects? Please clarify this point.
Reply: Thank you, honorable reviewer, for raising this important point. In this study, aging duration was treated as a fixed effect in the analysis to assess its impact on various meat quality characteristics. However, it was modeled as repeated measures to account for the correlation between observations collected at different aging durations from the same animal.
Comment: Is the description of the results and conclusions correct? Please must carefully check all results and make sure the accuracy of the description of results and conclusion. For example:
In conclusion, it’s stated that ‘’humpless bulls showed better instrumental tenderness and sensory characteristics’’, but from Table 5, there is no significant difference between Humped bulls and Humpless bulls; from Table 7, there is also no significant difference between two bull types, so how do you find those result and conclusion?
Reply: Thank you for emphasizing this important point. I have carefully reviewed both the results and conclusion, simplifying them to accurately reflect the study’s findings and ensure clearer comprehension for the readers.
The superscripts in the rows are denoted by capital letters (A-C), and in the columns, they are indicated by lowercase letters (a-c).
Tables 5 and 6 present the same results as outlined in the conclusion section. All muscles from humpless bulls exhibited lower WBSF values (indicating greater tenderness) compared to those from humped bulls. Similarly, all muscles from humpless bulls showed higher sensory scores (indicating better sensory characteristics) than those from humped bulls.
Additionally, I have highlighted the relevant sections of both tables for the reviewer’s convenience and clearer understanding.
Comment: Similarly, In conclusion, it’s stated that ‘’All meat quality characteristics were improved in 30±3 months than 21±2 months of age bulls’’. Why can you say this? I see from Table 5, this is no significant difference of cooking loss between two ages, and for WBSF, 30±3 months’ age bulls have higher value than those of 21±2 months.
Reply: Thank you for pointing out the mistake. I have revised the conclusion section to accurately reflect the study's findings, as suggested by the honorable reviewer. The updated content can be found in lines 462–464 of the manuscript and has been highlighted in red for your convenience.

Reviewer 2 Report (Previous Reviewer 3)
Comments and Suggestions for Authors
Dear authors,
thank you for your effort in improving your manuscript, which is now ready for publication. However, before publication, please look at one suggestion and a few notes about tipfelers, which is suggested to correct.
All the best

Author Response
Response to Reviewer # 2
Comment: Thank you for your effort in improving your manuscript, which is now ready for publication. However, before publication, please look at one suggestion and a few notes about tipfelers, which is suggested to correct.
Reply: Thank you, honorable reviewer, for appreciating the effort to improve the manuscript as per your suggestions. I have addressed all the points raised by you, and the revisions can be found in lines 22 and 138-143 of the updated manuscript.

This manuscript is a resubmission of an earlier submission. The following is a list of the peer review reports and author responses from that submission.
Round 1
Reviewer 1 Report
Comments and Suggestions for Authors
General comments:
The paper is interesting, but important information is missing in the materials and methods section, the statistical models adopted are not entirely clear, the paper is very complex to read, the results are not clearly reported in the Tables and sometimes they are not reported adequately in text. Please see the specific comments
Specific comments:
-Line 87-88: Why were the animals of different ages?
-Lines 92-93: Were the animals all slaughtered on the same day?
-Line 101: Please report at what position (e.g. rib level) the different samples were collected.
-Lines 93-111. What was the initial weight of the animals? Was it similar between humped and humpless? In my opinion, given the objective of the study more details on how the animals were raised should be reported. Were the animals raised individually or in pens? Too many details are missing. The authors report “The meat samples used in the current study were collected from the same animals as our previous study [16]”, but the work cited: Honikel 1987 does not seem to be relevant and if the paper [17] was intended, in this paper, if I'm not mistaken, 96 animals were considered.
-In my opinion the statistical model used is not correct (line 198), many variables considered (pH, color, WBSF) seem to have been repeated on samples from the same animals at different aging times, therefore the measurements carried out were not independent between them.
-Lines 203-204, it is not clear to me how the variables relating to the sensory analysis were analysed, it seems that only random effects were considered.
-Line 218: What do you mean by “better fattening”?
-Lines 220-221: “the pH of all muscles remained higher in 14 days, followed by 7 and 0 days of aging”, however this result is not clearly reported in Figure 1; only the effect of time within the muscle appears to be reported.
-Lines 277-278: It is not clear why the Authors are writing about incisors.
Lines 282-283: “In the present study, the drip loss of all muscles was higher at 14 days, followed by 7 and 0 days of aging”, I disagree, LT seems to have the same drip loss at 7 and 14 days.
Table 1 and others: I do not understand, it seems that the statistical model adopted is one (line 198), why are two standard errors reported? Considering that no interactions were significant, it is not clear why multiple comparisons were made within bull type - age - aging time.
Lines 245-246: this result is also very difficult to observe in the table, multiple comparisons were made only within days, the main effect of muscle is not evident.
Lines 294-296: It is unclear GM muscle had higher and lower cooking loss and WBSF in humpless than in humped, respectively. Conversely, for the other muscles a bull type effect was found for WBSF, but not for cooking loss.
Lines 302-303: “was linearly increased”, it is not appropriate, with the statistical model adopted this sentence is not supported by statistical analysis.
Line 322: what does “tenderness10” mean?
Lines 354-355: From Table 5, this sentence does not seem true.
Lines 361-362: “All the muscles showed higher ….. of aging” It does not seem always true (e.g. LT)
Lines 371-372: these results are not clearly reported in Table 5, the main effect of muscle is not evident
Lines 413-414: this result is also not clearly visible in the table
Conclusions: in the conclusions the main effect of muscle type is discussed, but in the tables this effect is not evident because in the column the muscles are compared by day.
Reviewer 2 Report
Comments and Suggestions for Authors
This paper explores the effects of different slaughter ages and ageing times on the quality of Bos indicus and Bos indicus × Bos taurus bull beef. Several typical muscle types are selected for study. The overall idea is clear, but there are still some points need to be modified:
1. L37, suggest changing “Bull Type” in Keywords to “Breed”.
2. L40-45, only colour and tenderness are mentioned in the Introduction, how about other quality attributes measured in this study?
3. Overall, the innovation of this study is not obvious. There have been many studies on the effects of age, ageing time, muscle type, and breed on bull beef quality. What is the difference of this study compared with others and what is the innovative point of this study? Additional explanation is needed.
4. L56 is“animal age”mentioned in this study“slaughter age”? Need to specify.
5. L87 18-24 months, 25-36 months are big ranges, and this expression is not accurate. Could you please specify the age?
6. L132 For meat colour measurement, L*, a*, b* are the basic three attributes, but there is b* missing, need to add b* value in result and discussion. Also, for C* saturation index, there is specific formulas to calculate it, what is the formulation?
7. How many replications are used in this study? Need to specify.
8. How about the interaction effect in this study?
9. L313 is the drip loss here correct? Or cooking loss?
10. TBARS was shown in this study, how about other oxidation or freshness parameters during preservation period? Also the basic composition of beef has not been detected in this study, how about intramuscular fat, protein, etc, content for these types of bull beef? It is suggested to supplement these experiments.
Comments on the Quality of English Language
Some words have inconsistent expressions, such as in L302, it showed cooked loss, while 303 showed cooking loss. The entire text needs to be checked and modified uniformly.
Reviewer 3 Report
Comments and Suggestions for Authors
Muscle-specific effect of age and aging time on meat quality characteristics of Bos indicus and Bos indicus × Bos taurus bulls
Review 1
General comments:
The manuscript investigates the physical chemical and sensory properties of four different muscles (Psoas major, Longissimus thoracis, Longissimus lumborum, and Gluteus Medius) of two different cattle breeds (Bos indicus and Bos indicus × Bos Taurus). The investigation includes different ages of the bulls (18–24 months and 25–36 months), and different postmortem meat aging times (0, 7, and 14 days). The research is interesting, current, and well-designed. However, additional explanations are needed, especially in research methods. Furthermore, an additional statistical analysis of the connection between individual sensory properties and other researched parameters is recommended.
I am not an expert in the English language, but I recommend proofreading.
Some specific comments are placed directly in the attached manuscript.

Round 2
Reviewer 3 Report
Comments and Suggestions for Authors
Muscle-specific effects of genotype, animal age, and wet aging duration on beef color, tenderness, and sensory characteristics
Review 2
The authors have made considerable effort to improve the article and have considered many of the recommendations from review 1. However, some suggestions and recommendations were not considered, and some may have been overlooked.
There are specific comments in the text of the attached article, and the authors are requested to read them carefully and consider their application.
I thank you in advance for your efforts.

Author Response
Response to Reviewer
Comment: age, .... tenderness,
Reply: Thank you for your comment. I have incorporated the suggested commas into the manuscript title as recommended by the respected reviewer.
Comment: The text in the Simple Summary should not contain overly technical terms such as Latin names. That's why it's simple. Perhaps (only in this Simple Summary) the Latin names of the muscles should be replaced with English names (eg Psoas major with tenderloin, etc.). Or, you can simply write that the research was done on three (or four?) different muscles (without naming them), and that the research showed differences (specify which ones). This is important because of the wider readership (please, read the Instructions for Authors). However, if the editors do not insist, I agree to leave it as it is.
Reply: Thank you for your guidance. The Latin names of the muscles have been replaced with their English equivalents in the revised manuscript, as reflected in lines 17-19.
Comment: The abreviation should be explanied. Put this abbreviation in brackets next to "lipid oxidation" in the line 28.
Reply: Thank you for your suggestion to improve the manuscript. The abbreviation has been added to the revised manuscript at lines 27-28.
Comment: It is recommended to put animal before the word age (to avoid misunderstandings).
Reply: Thank you for highlighting this important point. The word 'animal' has been added before 'age' throughout the manuscript.
Comment: What does LT mean?
Reply: Thank you for pointing this out. 'LT' stands for Longissimus thoracis, which was already abbreviated at line 56. To avoid any confusion, I have now included the full name and abbreviation at line 76 as well.
Comment: It is recommended to put "animal" before the word "age" (to avoid misunderstandings).
Reply: Thank you for highlighting this important point. The word 'animal' has been added before 'age' throughout the manuscript.
Comment: What does this sentence mean? You have to explain what kind of treatments it is, and what kind of replicas (before you write this sentence) or move this sentence to where you explain the treatments.
Reply: Thank you for pointing this out. The sentence has been removed from its inappropriate position.
Comment: displays
Reply: Thank you for your suggestion. The word 'display' has been changed to 'displays' in accordance with the recommendation of the honorable reviewer, as indicated in line 167 of the revised manuscript.
Comment: index
Reply: Thank you for pointing this out. The correction has been made and is noted at line 210 in the revised manuscript.
Comment: It is not clear whether a correlation was made or not. It is stated here as not determined, which implies that it was calculated, but not determined between any pair of investigated parameters, which is unlikely. Moreover, the text states several assumptions and conclusions based on a possible correlation (eg line 247 - negative correlation between TBAR and color indicators; also line 229). So, it is suggested again to calculate the correlation between the researched parameters (it's just one table). The table of coefficients of correlation can be placed in the supplement (or not), and the results can certainly be commented on in the text. In this way, making assumptions based on other research's results will be avoided.
Instead of correlation coefficients, PCA analysis can be done (even better).
Reply: Thank you for your valuable feedback. I have conducted Pearson correlation analysis among various meat quality characteristics, as suggested. The textural results have been elaborated more thoroughly with insights from this analysis. This addition helps to clarify the relationships between parameters and avoids relying solely on assumptions from other studies.
Comment: animal age
Reply: Thank you for highlighting this important point. The word 'animal' has been added before 'age' throughout the manuscript.
Comment: It is recommended to provide a reference. The pH of beef is usually higher in older animals for several reasons (structure of muscles and muscle fibers, lower level of glycogen, enzyme activity...).
Reply: Thank you for your concern. The sentence has been revised for a more thorough discussion of the results, and an appropriate reference has been added at line 230 in the revised manuscript, as suggested by the honorable reviewer.
Comment: If the correlation coefficient was calculated, then the impact would be known.
Reply: Thank you for your valuable feedback. I have conducted Pearson correlation analysis among various meat quality characteristics, as suggested. The impact of pH on meat L* value has now been elaborated more thoroughly, incorporating insights from this analysis in lines 254-258 of the revised manuscript.
Comment: bull type and age
Reply: Thank you for your suggestion. The sentence has been corrected as recommended by the esteemed reviewer and is now revised at line 311 in the manuscript.
Comment: It can be. But is there any reference to confirm this?
Reply: Thank you for pointing this out and for your suggestion. The reference has been added to the revised manuscript at line 318, as recommended by the honorable reviewer.
Comment: This is likely, but it would be good to provide a reference.
Reply: Thank you for pointing this out and for your suggestion. The reference has been added to the revised manuscript at line 325, as recommended by the honorable reviewer.
Comment: Given that this is an assumption, a reference should be provided.
Reply: Thank you for pointing this out and for your suggestion. The sentence has been revised and a reference has been added to the revised manuscript at line 329 as recommended by the honorable reviewer.
Comment: Given that this is an assumption, a reference should be provided.
Reply: Thank you for pointing this out and for your suggestion. The reference has been added to the revised manuscript at line 341, as recommended by the honorable reviewer.
Comment: bull type and age
Reply: Thank you for your suggestion. The sentence has been corrected as recommended by the esteemed reviewer and is now revised at line 340 in the manuscript.
Comment: Sensory attributes, such as tenderness and juiciness, are probably correlated with cooking loss and WBSF, and flavor is probably correlated with TBARS. It is highly recommended to calculate correlation coefficients (or PCA analysis).
Reply: Thank you for your valuable feedback. I conducted a Pearson correlation analysis among various meat quality characteristics, as suggested. Of the sensory characteristics, only tenderness exhibited a strong negative correlation with WBSF values. The lack of correlation between flavor and TBARS could be due to all the muscles maintaining TBARS values within the maximum acceptable limit of 2.0 mg MDA/kg throughout the experiment.
Comment: bull type and age,
Reply: Thank you for your suggestion. The sentence has been corrected as recommended by the esteemed reviewer and is now revised at line 416 in the manuscript.
Reply: It means that the WBSF should also be higher.
Comment: Yes, agree the WBSF also increases with age.
Comment: It may be, but marbling was not investigated in this study. So, at least back it up with a reference.
Reply: Thank you for your suggestion. A reference has been added to the revised manuscript at line 444, as recommended by the honorable reviewer.
Comment: animal age or bull age
Reply: Thank you for your suggestion. The sentence has been corrected as recommended by the esteemed reviewer and is now revised at line 449 in the manuscript.
Comment: What does this mean? If it is about the age of the bulls (months), then it does not match the one from the M&M chapter and what is written in the text, and the tables (21+2; 30+3).
Reply: Thank you for highlighting the important mistake. The age has been corrected in Figures 1 and 2.
Comment: type and age; apply to the other tables.
Reply: Thank you for your suggestion. The correction has been applied to all tables.
Comment: bull age
Reply: Thank you for your suggestion. The correction has been applied to all tables.
Comment: You can move the word "Bull" to the middle (between the words "type" and "age") and underline it, and write "type" and "age" below. This means that "type" and "age" refer to bulls
Reply: Thank you for your suggestion. The correction has been applied to all tables.

Round 3
Reviewer 3 Report
Comments and Suggestions for Authors
Dear authors,
thank you for your efforts in improving the quality of your article, which thus met the criteria for publication in the Animals journal.